# In Vivo Imaging of Acute Hindlimb Ischaemia in Rat Model: A Pre-Clinical PET Study

**DOI:** 10.3390/pharmaceutics16040542

**Published:** 2024-04-15

**Authors:** Gergely Farkasinszky, Judit Szabó Péliné, Péter Károlyi, Szilvia Rácz, Noémi Dénes, Tamás Papp, József Király, Zsuzsanna Szabo, István Kertész, Gábor Mező, Gabor Halmos, Zita Képes, György Trencsényi

**Affiliations:** 1Division of Nuclear Medicine and Translational Imaging, Department of Medical Imaging, Faculty of Medicine, University of Debrecen, H-4032 Debrecen, Hungarytrencsenyi.gyorgy@med.unideb.hu (G.T.); 2Gyula Petrányi Doctoral School of Allergy and Clinical Immunology, Faculty of Medicine, University of Debrecen, H-4032 Debrecen, Hungary; 3Doctoral School of Neuroscience, Faculty of Medicine, University of Debrecen, Nagyerdei St. 98, H-4032 Debrecen, Hungary; 4Division of Radiology and Imaging Science, Department of Medical Imaging, Faculty of Medicine, University of Debrecen, Nagyerdei St. 98, H-4032 Debrecen, Hungary; 5Division of Radiology, Department of Medical Imaging, Faculty of Medicine, University of Debrecen, H-4032 Debrecen, Hungary; 6Department of Biopharmacy, Faculty of Pharmacy, University of Debrecen, H-4032 Debrecen, Hungary; 7Institute of Chemistry, Faculty of Science, Eötvös Loránd University, H-1053 Budapest, Hungary; 8MTA-ELTE, Research Group of Peptide Chemistry, Hungarian Academy of Sciences, Eötvös L. University, H-1053 Budapest, Hungary

**Keywords:** aminopeptidase N (APN/CD13), angiogenesis, 2-[^18^F]FDG, [^68^Ga]Ga-NOTA-c(NGR), ischaemia, Peripheral Arterial Disease (PAD), positron emission tomography (PET), preclinical

## Abstract

Background: To better understand ischaemia-related molecular alterations, temporal changes in angiogenic Aminopeptidase N (APN/CD13) expression and glucose metabolism were assessed with PET using a rat model of peripheral arterial disease (PAD). Methods: The mechanical occlusion of the base of the left hindlimb triggered using a tourniquet was applied to establish the ischaemia/reperfusion injury model in Fischer-344 rats. 2-[^18^F]FDG and [^68^Ga]Ga-NOTA-c(NGR) PET imaging performed 1, 3, 5, 7, and 10 days post-ischaemia induction was followed by Western blotting and immunohistochemical staining for APN/CD13 in ischaemic and control muscle tissue extracts. Results: Due to a cellular adaptation to hypoxia, a gradual increase in [^68^Ga]Ga-NOTA-c(NGR) and 2-[^18^F]FDG uptake was observed from post-intervention day 1 to 7 in the ischaemic hindlimbs, which was followed by a drop on day 10. Conforming pronounced angiogenic recovery, the NGR accretion of the ischaemic extremities differed significantly from the controls 5, 7, and 10 days after ischaemia induction (*p* ≤ 0.05), which correlated with the Western blot and immunohistochemical results. No remarkable radioactivity was depicted between the normally perfused hindlimbs of either the ischaemic or the control groups. Conclusions: The PET-based longitudinal assessment of angiogenesis-associated APN/CD13 expression and glucose metabolism during ischaemia may continue to broaden our knowledge on the pathophysiology of PAD.

## 1. Introduction

Peripheral artery disease (PAD) is a common condition characterized by the narrowing or the blockage of the arteries, typically in the lower limbs, that leads to reduced blood flow and subsequent tissue ischaemia [1,2,3]. Given that such diseases represent a major health concern worldwide, an increasing number of studies have turned towards the profound investigation of the dynamic changes in tissue perfusion during acute hindlimb ischaemia (AHI) [4,5,6]. In this respect, pre-clinical imaging techniques seem to be promising tools to shed light on AHI-related pathophysiological processes, which may eventually open an avenue for the development of effective diagnostic and therapeutic strategies [7,8].

In vivo imaging techniques, including positron emission tomography (PET), offer a valuable advantage to evaluate biological processes in real-time, at the cellular or the molecular level, and even in a quantitative way [9,10]. Data from the literature have already confirmed the role of PET imaging in tracking AHI-associated angiogenic alterations by applying experimental models of AHI [11].

Intensive interest has recently been centered around the investigation of one of the most important angiogenic biomarkers, aminopeptidase N (APN/CD13)-a Zn^2+^-dependent transmembrane ectopeptidase, which has a major contribution to angiogenesis as well as neoangiogenesis [12,13,14]. Both normal and activated blood vessel endothelial cells are reported to abundantly express APN/CD13 [15,16]. Based on phage display technology results, peptides with an asparagine–glycine–arginine (NGR) motif specifically bind to APN/CD13 [17]. NGR compounds labeled with different radioisotopes, such as Copper-64 (^64^Cu), Gallium-68 (^68^Ga), Technetium-99m (^99m^Tc), Lutetium-177 (^177^Lu), or Rhenium-188 (^188^Re), proved to be highly valuable in recognizing APN/CD13 under pre-clinical circumstances [18,19,20,21,22,23].

Due to the outstanding physical features (T_1/2_: 67.71 min), decay characteristics (Eβ^+^_average_: 830 KeV, maximum β^+^ energy: 1.92 MeV, Iβ^+^: 89%, Eγ: 1077 KeV, Iγ: 3.2%) and seamless generator-based production of ^68^Ga, it has recently become one of the most attractive radiometals for NGR labelling purposes [24,25,26,27,28,29]. Using various experimental tumor models, ^68^Ga-labeled NGR vectors appeared to be successful in the visualization of cancer-linked angiogenesis and APN/CD13 overexpressing malignancies [18,24,25]. Beyond oncological applications, however, our previous study with a surgically induced diabetic retinopathy (DR) rat model strengthened the applicability of the ^68^Ga-appended NGR sequence (c(NGR)) in perfusion imaging as well [30].

Aside from radiolabeled NGR derivatives, PET imaging with 2-[^18^F]FDG—the key marker of glucose metabolism—may also provide a useful means to gain a deeper insight into the underlying mechanisms behind ischaemia and reperfusion [31].

Herein, PET imaging was used to explore the feasibility of [^68^Ga]Ga-NOTA-c(NGR) in the assessment of ischaemia-induced APN/CD13 changes in a preclinical ischaemic hindlimb model resembling human PAD. Additionally, 2-[^18^F]FDG PET was performed to assess ischaemia-related metabolic changes in the same model.

## 2. Materials and Methods

### 2.1. PET Tracers

Based on the method of Mikecz et al., 2-[^18^F]FDG is routinely synthesized in the radiochemical laboratory of the Division of Nuclear Medicine and Translational Imaging, Department of Medical Imaging, Faculty of Medicine, University of Debrecen (Debrecen, Hungary) for human PET diagnostic purposes in compliance with the Good Manufacturing Practice (GMP) [32]. It was obtained from a fully automated GE Tracerlab FX_FDG_ module (GE Healthcare, North Richland Hills, TX, USA) [33]. The APN/CD13-specific [^68^Ga]Ga-NOTA-c(NGR) (Figure 1) was synthesized manually using a pharmaceutical grade Germanium-68/Gallium-68 (^68^Ge/^68^Ga) generator system (Eckert & Ziegler, Berlin, Germany) according to the manufacturer’s recommendation [18].

### 2.2. Animal Models

Sixteen-week-old male Fischer-344 rats (n = 15; Animalab Ltd., Budapest, Hungary) ranging in weight from 250 to 300 g were studied. Animals were kept under conventional laboratory circumstances in individually ventilated cages (IVC) (number of animals/cages = 2/1) where the temperature was maintained at 21 ± 2 °C with a relative humidity of 55 ± 10% throughout the whole study. Free access to water and semi-synthetic rodent food (VRF-1 SDS) was provided for all study animals. Artificial light/dark schedule of 12 h was ensured.

All procedures used in the present research were authorized by the Institutional Animal Welfare Committee of the University of Debrecen, Debrecen, Hungary (permission registration Nr.: 28/2022/DEMÁB) and were in accordance with all applicable sections of the Hungarian Laws as well as the directions and regulations of the European Union.

### 2.3. Rat Model of Acute Hindlimb Ischaemia

Ischaemia-reperfusion (I/R) was recapitulated on the basis of the previously reported method [34]. Briefly, unilateral hindlimb ischaemia was induced through the clamping of the base of the left hindlimb using a tourniquet. The cyanotic discolouration of the affected limb and pulse oximeter measurements confirmed the arterial occlusion and related ischaemia [35]. After a 120 min long ischemic period, the tourniquet was removed.

The rats were randomly categorized into the following groups: control (sham-operated) group (n = 5) and ischaemic group (n = 10). Based on the presence or the absence of ischaemia, the hindlimbs of the tourniquet-induced ischaemic group were subdivided into I/R subgroup (left hindlimb), where the tourniquet technique was applied; and non-I/R subcategory, which indicated intact perfusion in the right lower extremity. As for the control cohort, both hindlimbs (left and right) belonged to the non-I/R subgroup.

### 2.4. Small Animal PET Imaging

As part of in vivo imaging 10.03 ± 2.69 MBq of [^68^Ga]Ga-NOTA-c(NGR) and four hours later 11.33 ± 1.71 MBq of 2-[^18^F]FDG were intravenously injected into both the ischaemic and the control rats via the lateral tail vein 1, 3, 5, 7, and 10 days post-I/R induction. Sixty minutes post-tracer administration, 20 min static PET images were acquired on the hindlimb region of the rats under isoflurane-induced anaesthesia (3% isoflurane (Forane), AbbVie, Budapest, Hungary; OGYI-T-1414/01) using the MiniPET II device of the Division of Nuclear Medicine and Translational Imaging, Department of Medical Imaging, Faculty of Medicine, University of Debrecen, Hungary.

### 2.5. PET Data Analysis

BrainCad image analysis software (Version: 1.124) was employed for quantitative PET data assessment (University of Debrecen, Debrecen, Hungary). VOIs (volume of interests) were manually placed around the region of the hindlimb, and using the subsequent formula standardized uptake values (SUV) were determined to define the radiotracer uptake within the concerned VOI: SUV = [VOI activity (Bq/mL)]/[injected activity (Bq)/animal weight (g)], assuming a density of 1 g/mL.

### 2.6. Western Blot Analysis

To assess the APN/CD13 receptor profile of the extracted tissue samples Western blot analysis was accomplished. After removal, the hindlimb tissue samples were immediately frozen in liquid nitrogen and stored at −80 °C until further processing.

For tissue lysate preparation, samples were suspended and lysed in ice-cold M-PER protein lysis buffer (Thermo Fisher Scientific, Waltham, MA, USA) supplemented with protease and phosphatase inhibitors (SigmaAldrich, St. Louis, MO, USA). Homogenization was performed using Tissue Ruptor (IKA^®^-WERKE GmbH, Staufen im Breisgau, Germany). The protein quantification of the cell lysate was carried out with the application of homemade Bradford reagent. All samples were diluted with 4× Laemmli buffer. Equal amounts (40 µg) of each protein in equal volumes were boiled at 95 °C for 8 min. Corresponding to the molecular weight of the target protein, the lysates were separated on 10% Sodium Dodecyl Sulfate (SDS) polyacrylamide gel using electrophoresis (SDS-PAGE) and electrotransferred onto a polyvinylidene fluoride (PVDF) membrane (Millipore, Burlington, MA, USA). Precision Plus Protein Dual Color Standard (BioRad Laboratories, Irvine, CA, USA) was used as a molecular weight marker. After blocking with 5% milk-TBS-Tween, the membranes were incubated with primary mouse anti-rat APN/CD13 antibody at a dilution of 1:500 overnight at 4 °C (sc-13536, Santa Cruz Biotechnology Inc., Dallas, TX, USA). Following the incubation with the primary antibodies, the membranes were washed and probed with HRP-tagged anti-mouse IgG secondary antibody (1:2000 dilution, Thermo Fisher Scientific, Waltham, MA, USA). Enhanced chemiluminescence reaction was used to detect the antibody-labeled bands. The intensity of each band was normalized to HPRT (anti-HPRT, 1:1000 HPRT Antibody (B-11) mouse monoclonal, sc-393901; Cell Signaling Technology, Danvers, MA, USA). ChemiDoc Imaging System (Bio-Rad, Hercules, CA, USA) was used for both the membrane visualization and densitometry (rat kidney tissue was used for positive APN/CD13 protein expression as a positive control) [36].

### 2.7. Immunohistochemical Analysis

Immunostaining was performed as described previously [37]. Briefly, representative hindlimb muscle tissue specimens from each experimental animal were stained with a monoclonal mouse anti-rat CD13 antibody (RTU; Proteintech, Rosemont, IL, USA) on an IHCeasy CD13 Ready-To-Use IHC Kit^®^ (Proteintech, Rosemont, IL, USA). Immunohistochemical analyses (IHC) was performed on formalin-fixed, frozen, whole-tissue sections using a polymer-based IHC method with appropriate positive and negative controls. Photos were taken applying Nikon Eclipse E800 (Nikon Corporation, Tokyo, Japan) microscope.

### 2.8. Statistical Analysis

A commercial software package (MedCalc 18.5, MedCalc Software, Mariakerke, Belgium) was used for all statistical analyses. Student’s two-tailed *t*-test, two-way ANOVA, and Mann–Whitney rank-sum tests were applied to determine the significance. The significance level was set at *p* ≤ 0.05. All data are presented as mean ± SD of at least three independent series of measurements.

## 3. Results and Discussion

### 3.1. In Vivo Longitudinal Study of APN/CD13 Expression with [^68^Ga]Ga-NOTA-c(NGR) in Rat Models of Acute hindlimb Ischaemia and in Healthy Counterparts

The [^68^Ga]Ga-NOTA-c(NGR) PET scans of the I/R hindlimbs were compared to those of the non-I/R hindlimbs of the tourniquet cohort and the healthy hindlimbs of the control group (non-I/R) to explore ischaemia-associated effects on the alterations in the APN/CD13 receptor pattern.

The radiopharmaceutical uptake of the hindlimbs on [^68^Ga]Ga-NOTA-c(NGR) PET is demonstrated in Figure 2A,B and Figure 4A. Upon visual assessment of the decay-corrected PET images, the uptake of the I/R hindlimb of the ischaemic group showed a gradual increase after ischaemia induction from day 1 until day 7, and then on the 10th day a sharp decrease in [^68^Ga]Ga-NOTA-c(NGR) was observed. On the contrary, faint radioactivity could be depicted in the non-I/R hindlimb of the ischaemic group and in both hindlimbs (non-I/R) of the control group (Figure 2 and Figure 4).

Figure 2C,D present the quantitative assessment of [^68^Ga]Ga-NOTA-c(NGR) accumulation. In accordance with the visual analysis, a continuous increase in the SUVmean values for the I/R hindlimbs in the ischaemic group could be observed from the 1st day (SUVmean: 0.090 ± 0.005) until the 7th day (SUVmean: 0.230 ± 0.015). Since data from the literature stated that the cellular adaptation to hypoxia/anoxia is regulated by hypoxia inducible factors, including Hypoxia Inducible Factor-1α (HIF-1α), which triggers the transcription of various pro-angiogenic genes and the expression of related proteins [38,39], we suppose that the experienced radiopharmaceutical uptake increment could be attributed to the elevation of proangiogenic factors, such as APN/CD13, induced by ischaemia-mediated hypoxia.

Similarly to the qualitative assessment, the initial uptake increment with the highest values measured on day 7 (SUVmean: 0.230 ± 0.015) was followed by a sharp drop 10 days after the tourniquet-induced ischaemia (SUVmean: 0.123 ± 0.010). Nevertheless, while the exact reason behind this uptake trend remains to be elucidated, we suppose that the angiogenic biomarkers are more likely to be overexpressed at the onset of ischaemia to compensate for the altered blood supply and hypoxia-related changes. Using Copper-64 (^64^Cu)-labeled vascular endothelial growth factor-121 ([^64^Cu]Cu-VEGF-121) for angiogenesis imaging of hindlimb ischaemia, Willmann and co-workers also detected the most noticeable perfusion recovery during the first week after surgery (with the peak on the 8th day, and then decreasing uptake) [40].

Identically, in a prior study by Kobayashi et al. with a myocardial infarction (MI) model of C57BL/6J mice, prominent expression of proangiogenic A (VEGF-A) was reported in low oxygenated myocardial regions 24 h after infarction induction [41].

On the contrary, however, other findings suggest that the early phases of ischaemia favor antiangiogenic mechanisms rather than vasculogenesis [42].

Five, seven, and ten days post ischaemia generation, significantly elevated [^68^Ga]Ga-NOTA-c(NGR) accretion was registered in the I/R hindlimbs of the ischaemic group compared to the normally perfused (non-I/R) hindlimbs of the same cohort (*p* ≤ 0.05), with SUVmean values of 0.138 ± 0.010, 0.230 ± 0.015, and 0.123 ± 0.010 for the I/R hindlimbs on the 5th, 7th, and 10th days, respectively, and 0.085 ± 0.012, 0.088 ± 0.015, and 0.093 ± 0.011 for the non-I/R hindlimbs at the same measurement time points (Figure 2C,D). The maximal [^68^Ga]Ga-NOTA-c(NGR) uptake value was recorded on day 7, which was similar to the findings of Almutairi et al., who tested the feasibility of a Bromine-76 (^76^Br)-labeled RDG-based dendritic nanoprobe targeting α_v_β_3_ for angiogenesis imaging using an ischaemic mouse hindlimb model, and registered the highest radioactivity a week after the intervention [43]. In a similar manner, cyclic RGD peptide [c(RGDyK)] labeled with ^68^Ga and ^99m^Tc-labeled RGD containing NC100692 ([^99m^Tc]Tc-NC100692) were also reported to show considerable accumulation in ischaemic murine hindlimbs 7 days after femoral artery occlusion [44,45].

Currently, no considerable disproportion could be detected between the I/R and the non-I/R hindlimbs of the ischaemic animals regarding the SUVmax values.

Nevertheless, on the 1st and 3rd days after clamping, no remarkable difference could be detected between the SUVmean figures of the I/R and non-I/R hindlimbs of the ischaemic group (SUVmean values: 0.090 ± 0.005 and 0.100 ± 0.007 on the 1st and 3rd days, respectively, SUVmean values: 0.080 ± 0.005 and 0.075 ± 0.006 on the 1st and 3rd days, respectively; *p* ≤ 0.05). In addition, on the 5th and 7th investigation time point (day 5 and day 7), the I/R hindlimbs showed approximately 2-to-3-fold higher [^68^Ga]Ga-NOTA-c(NGR) uptake in comparison with that of the normally perfused (non-I/R) ones of the ischaemic group (Figure 2C,D). Furthermore, the radiopharmaceutical concentration of the I/R hindlimbs (SUVmean: 0.138 ± 0.010 (5th day) and 0.230 ± 0.015 (7th day)) was approximately twice or three times as much as that of either hindlimbs (non-I/R) of the healthy animal group at the same measurement times (left hindlimb SUVmean: 0.070 ± 0.004 (5th day) and 0.073 ± 0.008 (7th day); right hindlimb SUVmean: 0.070 ± 0.005 (5th day) and 0.071 ± 0.009 (7th day)). Using a diabetic retinopathy mouse model, Farkasinszky and colleagues reported similar observations with the same radiopharmaceutical to that of ours [30].

Additionally, we did not find any remarkable differences between the NGR tracer uptake of the non-I/R hindlimbs of the ischaemic group and that of the non-I/R hindlimbs of the control cohort (*p* ≤ 0.05), nor between the left and right non-I/R hindlimbs of the control group (*p* ≤ 0.05).

Overall, the elevated [^68^Ga]Ga-NOTA-c(NGR) accumulation registered in the ischaemic hindlimbs of the study animals highlights the crucial role of APN/CD13 in the ischaemia-induced neo-angiogenesis related to PAD. The longitudinal assessment of the APN/CD13 expression pattern during ischaemia using PET may continue to broaden our knowledge on the pathology of PAD, contributing not only to the discovery of novel therapeutic targets, but also granting an opportunity to evaluate their therapeutic efficacy prior to human application.

### 3.2. In Vivo Longitudinal Assessment of Ischaemic and Normally Perfused Hindlimb Metabolism with 2-[^18^F]FDG

Since the efficacy of 2-[^18^F]FDG micro-imaging in the assessment of cardiac or cerebral ischaemia was exemplified in earlier preclinical studies [46,47,48], herein, an ischaemic hindlimb rat model resembling PAD was subjected to miniPET examinations to test the feasibility of the radiolabeled glucose analogue in the follow-up of ischaemia-associated dynamic changes.

The decay-corrected transaxial 2-[^18^F]FDG PET images and the quantitative uptake parameters are shown in Figure 3A–D and Figure 4B. Confirmed using a quantitative PET data assessment, a steady increase in 2-[^18^F]FDG accumulation can be visualized (Figure 3) in the I/R hindlimbs of the ischaemic group from the 1st (SUVmean: 0.70 ± 0.26) until the 7th (SUVmean: 1.18 ± 0.32) investigation time points, which was followed by a decline on the 10th day (SUVmean: 0.59 ± 0.30), and this uptake trend was in line with that of observed in the case of [^68^Ga]Ga-NOTA-c(NGR). In addition, the 2-[^18^F]FDG concentration of the I/R hindlimbs registered on days 3 (SUVmean: 0.90 ± 0.28), 5 (SUVmean: 1.09 ± 0.30), and 7 (SUVmean: 1.18 ± 0.32) was remarkably higher (*p* ≤ 0.05) in comparison with the uptake of the non-I/R hindlimbs of the ischaemic group and those of either non-I/R hindlimbs of the healthy animals measured on the same experimental days (Figure 3); moreover, the I/R hindlimbs presented approximately 2-to-3-times higher radioactivity relative to that of the non-I/R hindlimbs of the same animal population.

Taken together with the findings in the literature to date, we suggest that the aggravated metabolic activity experienced in the I/R hindlimbs could be a physiological inflammatory response reaction to ischaemia-associated tissue hypoperfusion. As part of ischaemia-linked inflammation, an increasing number of macrophages are gathered into the ischaemic tissue [49,50], which, due to their high glucose needs [51], could be responsible for the elevated metabolism and subsequent high 2-[^18^F]FDG accumulation. However, post-intervention inflammation caused by tourniquet-induced mechanical occlusion can lead to increased non-specific 2-[^18^F]FDG uptake, which must be addressed during the overall assessment of 2-[^18^F]FDG distribution.

Neither on the 1st nor on the 10th day post-ischaemia induction did we encounter any significant disparities between the metabolic activity of the I/R and the non-I/R hindlimbs of the ischaemic group, with the respective SUVmean values being 0.70 ± 0.26 and 0.65 ± 0.21 on the 1st day and 0.59 ± 0.30 and 0.51 ± 0.17 on the 10th day (*p* ≤ 0.05). Similarly, the 2-[^18^F]FDG uptake of the I/R hindlimbs did not differ from the radioactivity of the non-I/R hindlimbs of the control cohort 1 and 10 days after the intervention (*p* ≤ 0.05). The quantitative PET parameters of the corresponding groups are exhibited in Figure 3C,D.

Identically to the observations for the c(NGR) scans, neither the non-I/R hindlimbs of the ischaemic group nor the non-I/R hindlimbs of the controls displayed meaningful metabolic activity; moreover, their metabolism did not show any difference from each other (*p* ≤ 0.05), which was also in accordance with the results for the NGR uptake.

Considering the current findings, 2-[^18^F]FDG PET imaging could provide complementary information on ischaemia-related molecular changes; therefore, it could be a viable alternative to NGR PET in the molecular imaging of the neo/angiogenic processes in PAD. Aiming to overcome the limitations of 2-[^18^F]FDG PET derived from non-specific tracer accumulation, however, seems to be a part of future work.

### 3.3. Western Blot Analysis

Western blot-based APN/CD13 protein analysis was executed in both animal cohorts to make correlations with the in vivo uptake trend of [^68^Ga]Ga-NOTA-cNGR. Corresponding to the NGR PET data, a significantly higher amount of APN/CD13 protein was detected in the I/R hindlimbs of the ischaemic group relative to that of the non-I/R hindlimbs of either the ischaemic or the control cohort (*p* ≤ 0.05) on the 5th, 7th, and 10th day post-ischaemia induction (Figure 5). As expected on the basis of the NGR uptake, no meaningful difference was detected between the protein expression of the non-ischaemic hindlimbs of the ischaemic group (non-I/R) and that of the controls (non-I/R). The 10-day Western blot findings on the diabetic retinopathy rat model established by Farkasinszky et al. were consistent with the present results [30]. The drop in protein expression observed at day 10, subsequent to the initial increase from day 1 to day 7, may stem from various factors inherent to the dynamic nature of neo-angiogenesis and tissue repair following ischemic injury. One plausible explanation pertains to the resolution of the acute inflammatory response typically associated with tissue injury and ischaemia. As the initial insult wanes, there could be a diminution in the expression of pro-inflammatory mediators, leading to reduced recruitment and activation of inflammatory cells, notably macrophages expressing APN/CD13 [52]. Although the radiotracer uptake did not differ significantly between the I/R hindlimbs and the non-I/R ones (both the ischaemic and the control group) on day 1 and day 3, a remarkable distinction was observed regarding their APN/CD13 protein expression on the same experimental days (*p* ≤ 0.05). We suppose that although the ischaemia-induced changes were already manifested at the receptorial level, the difference between the extent of the receptor expression of the ischaemic and non-ischaemic groups was not yet so pronounced as to cause a notable difference in the radiopharmaceutical uptake values. Figure 5 displays the results of Western blotting.

### 3.4. Immunohistochemical Analyses of APN/CD13 Expression

Hindlimb muscle tissue extracts taken from the I/R hindlimbs of the ischaemic group were immunostained to follow the ischaemia-associated temporal changes in APN/CD13 expression at the molecular level. For the comparison of the receptorial pattern of the ischaemic and normally perfused skeletal muscle tissues, samples removed from the non-I/R hindlimbs of the ischaemic group and from those of the control cohort were also analyzed. As indicated in Figure 6, a steady increase could be registered in the APN/CD13 expression (brown staining) of the ischaemic muscle tissue specimens (I/R) from day 1 (Figure 6A) until day 7 (Figure 6E), which was followed by a slight decrease on the 10th day after the intervention, and this receptorial profile was in accordance with the experienced [^68^Ga]Ga-NOTA-c(NGR) uptake trend visualized on the corresponding PET scans, as well as with the results of the Western blot analysis at the same investigation time points.

On the contrary, hardly any APN/CD13 positivity was found either on the histological sections of the normally perfused muscle tissues of the ischaemic group (non-I/R; Figure 6F) or those of the control group (non-I/R; Figure 6G), which also correlated with the NGR PET findings. Based on the tendency of the upregulation of APN/CD13 expression, it seems that the angiogenic recovery is most pronounced within the first days after the beginning of ischaemia.

Particular attention must be paid to the study limitations arising from the utilization of a small number of animals in the research. In our study, the use of a limited number of animals, albeit common in preclinical research, may result in restrictions on the generalizability of our findings to human populations. Additionally, the translation of these findings to human subjects is required to be carefully considered due to inherent inter-species differences. Furthermore, a notable limitation was presented by the absence of comprehensive immunological panels, as valuable insights into the underlying mechanisms driving observed outcomes could be offered by such assays. Moreover, valuable methods are available for the in-depth in vivo evaluation of angiogenesis, which could be explored in future studies. Priority should be given in future investigations to the inclusion of a broader array of immunological assays to further elucidate the complex interactions between the experimental interventions and the immune system. Despite these limitations, valuable preliminary data were contributed to the field by our study, underscoring the importance of subsequent research efforts aimed at addressing these constraints and expanding upon our findings.

## 4. Conclusions

On the whole, the efficacy of [^68^Ga]Ga-NOTA-c(NGR) in monitoring ischaemia-related temporal changes in APN/CD13 expression coupled with the favorable physical characteristics of ^68^Ga, [^68^Ga]Ga-NOTA-c(NGR) targeted at APN/CD13 is a potential candidate to be added to the diagnostic armamentarium of angiogenesis imaging. A better understanding of ischaemia-mediated angiogenic processes using the non-invasive PET technique along with molecular diagnostics may have future therapeutic implications in human patient care. In addition, given the similarity between the pathophysiological processes of PAD and other cardiovascular comorbidities, such as generalized atherosclerosis, coronary heart disease, and myocardial infarction, [^68^Ga]Ga-NOTA-c(NGR) may be groundbreaking in the follow-up of the angiogenesis of these diseases both at the pre-clinical and the clinical level.

## Figures and Tables

**Figure 1 pharmaceutics-16-00542-f001:**
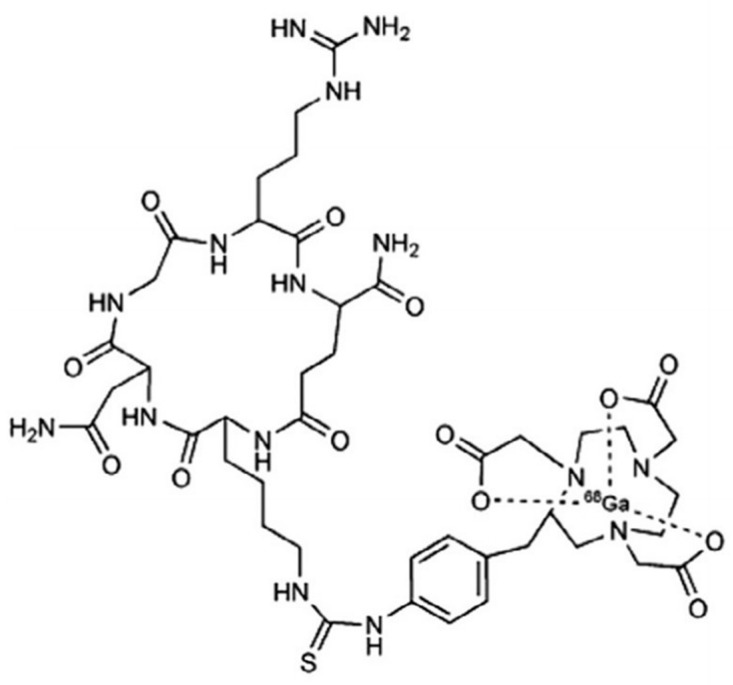
Chemical structure of [68Ga]Ga-NOTA-c(NGR). [^68^Ga]Ga: gallium-68; c(NGR): cyclic NGR; NGR: asparagine–glycine–arginine; NOTA: 1,4,7-triazacyclononane-triacetic acid.

**Figure 2 pharmaceutics-16-00542-f002:**
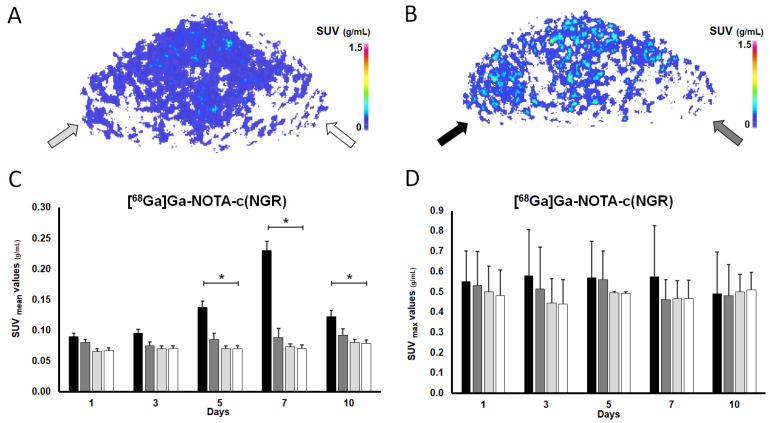
In vivo longitudinal evaluation of APN/CD13 expression using [^68^Ga]Ga-NOTA-c(NGR) PET imaging. Representative, decay-corrected transaxial PET images of the (**A**) control and the (**B**) ischaemic group 7 days after tourniquet-induced hindlimb ischaemia and 60 min post-intravenous injection of 10.03 ± 2.69 MBq of [^68^Ga]Ga-NOTA-c(NGR). Quantification of [^68^Ga]Ga-NOTA-c(NGR) uptake (**C**,**D**) 1, 3, 5, 7, and 10 days after ischaemia was induced and 60 min after the intravenous administration of the radiotracer. Values are expressed in SUV_mean_ (**C**) and SUV_max_ (**D**). Asterisk indicates significance: * *p* ≤ 0.05. Light grey arrow and column show the normally perfused hindlimb of the control group (non-I/R, **left**), n = 5; white arrow and column the normally perfused hindlimb of the control group (non-I/R, **right**), n = 5; black arrow and column the ischaemic (I/R) hindlimb of the ischaemic group (**left**), n = 10; and grey arrow and column show the normally perfused hindlimb of the ischaemic group (non-I/R, **right**), n = 10. APN/CD13: aminopeptidase N; c(NGR): cyclic NGR; [^68^Ga]Ga: gallium-68; I/R: ischaemia-reperfusion; NGR: asparagine–glycine–arginine; NOTA: 1,4,7-triazacyclononane-triacetic acid; PET: positron emission tomography; SUV: standardized uptake value.

**Figure 3 pharmaceutics-16-00542-f003:**
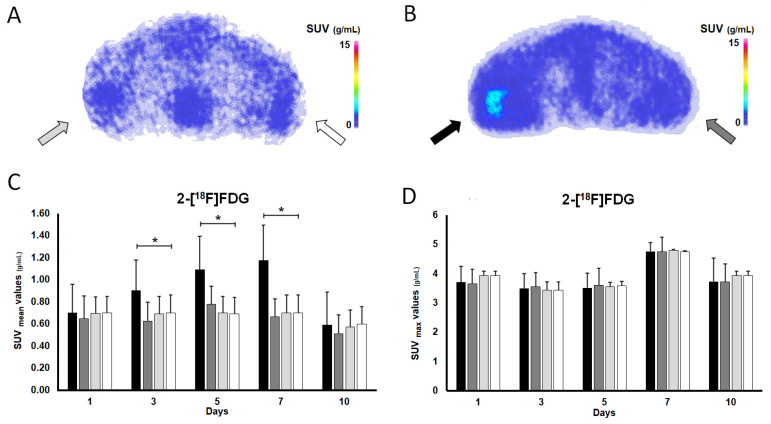
In vivo longitudinal assessment of metabolic changes in ischaemic and normally perfused hindlimbs with 2-[^18^F]FDG PET imaging. Representative, decay-corrected transaxial PET images of the (**A**) control and the (**B**) ischaemic group 7 days after tourniquet-induced hindlimb ischaemia and 60 min post-intravenous injection of 11.33 ± 1.71 MBq of 2-[^18^F]FDG. Quantification of 2-[^18^F]FDG uptake (**C**,**D**) 1, 3, 5, 7, and 10 days after ischaemia was induced and 60 min after the intravenous administration of the radiotracer. Values are expressed in SUV_mean_ (**C**) and SUV_max_ (**D**). Asterisk indicates significance: * *p* ≤ 0.05. Light grey arrow and column show the normally perfused hindlimb of the control group (non-I/R, **left**), n = 5; white arrow and column the normally perfused hindlimb of the control group (non-I/R, **right**), n = 5; black arrow and column the ischaemic (I/R) hindlimb of the ischaemic group (**left**), n = 10; and grey arrow and column show the normally perfused hindlimb of the ischaemic group (non-I/R, **right**), n = 10. I/R: ischaemia-reperfusion; 2-[^18^F]FDG: fluorine-18 fluorodeoxyglucose; PET: positron emission tomography; SUV: standardized uptake value.

**Figure 4 pharmaceutics-16-00542-f004:**
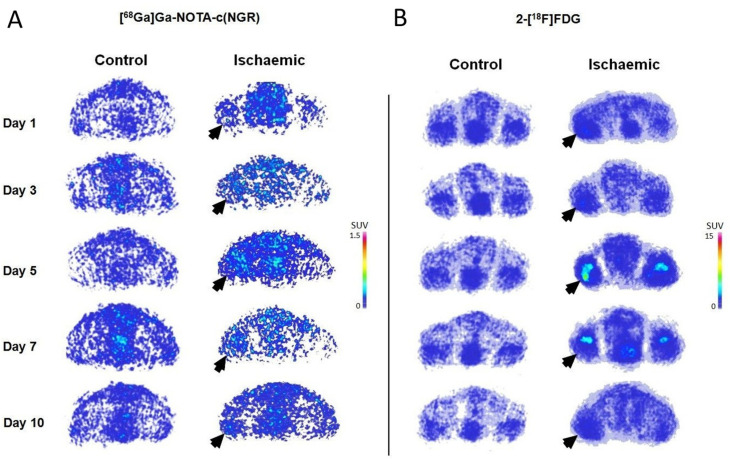
Longitudinal visual assessment of APN/CD13 expression (**A**) and the metabolism (**B**) and the temporal changes in I/R hindlimb rat model and healthy counterparts with PET technique. Representative transaxial images of in vivo PET results of control and ischaemic rats administered with 10.03 ± 2.69 MBq of [^68^Ga]Ga-NOTA-c(NGR) (**A**) and 11.33 ± 1.71 MBq of 2-[^18^F]FDG (**B**) and 1, 3, 5, 7, and 10 days after ischaemia induction are demonstrated. Visually, a gradual increase in [^68^Ga]Ga-NOTA-c(NGR) and 2-[^18^F]FDG can be seen in the ischaemic hindlimbs (I/R) of the ischaemic cohort from day 1 until day 7, which was followed by a decline 10 days after ischaemia was induced. Black arrows indicate the ischaemic (I/R) hindlimbs of the ischaemic group. APN/CD13: aminopeptidase N; c(NGR): cyclic NGR; 2-[^18^F]FDG: fluorine-18 fluorodeoxyglucose; [^68^Ga]Ga: gallium-68; I/R: ischaemia-reperfusion; NGR: asparagine–glycine–arginine; NOTA: 1,4,7-triazacyclononane-triacetic acid; PET: positron emission tomography; SUV: standardized uptake value.

**Figure 5 pharmaceutics-16-00542-f005:**
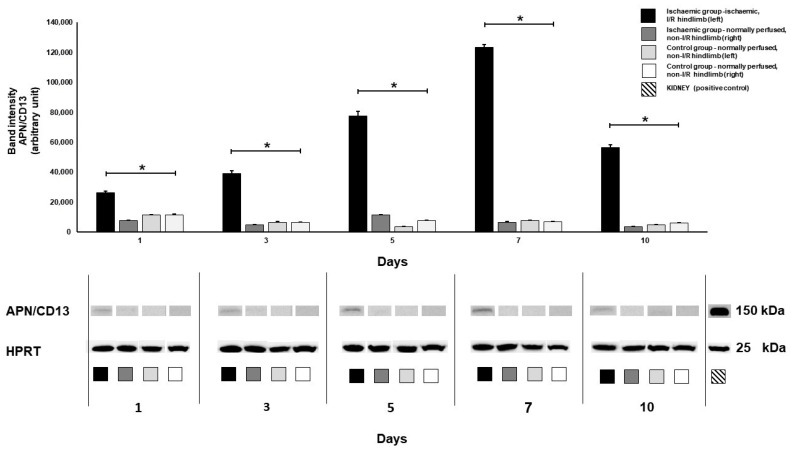
Western blot analysis of ischaemic (I/R) and normally perfused (non-I/R) hindlimb muscle tissue lysates for APN/CD13 expression (150 kDa) 1, 3, 5, 7, and 10 days post-ischaemia induction. HPRT (25 kDa) was used as a loading control. Rat kidney tissue specimens (striped) were the positive controls. The amount of APN/CD13 protein steadily increased in the I/R hindlimbs from day 1 until day 7, which was followed by a sharp drop 10 days after tourniquet-induced ischaemia. Significantly elevated APN/CD13 expression level was found in the I/R hindlimbs of the ischaemic group compared to the non-I/R hindlimbs of both the ischaemic and the control cohort (*p* ≤ 0.05). No difference could be depicted between the APN/CD13 receptorial pattern of the normally perfused hindlimbs (non-I/R) of the ischaemic and the control group. Black: ischaemic group ischaemic (I/R) hindlimb (left), n = 10; grey: ischaemic group normally perfused (non-I/R) hindlimb (right), n = 10; light grey: control group normally perfused (non-I/R) hindlimb (left), n = 5; white: control group normally perfused (non-I/R) hindlimb (right), n = 5. Values are expressed in arbitrary units. Asterisk indicates significance: * *p* ≤ 0.05. APN/CD13: aminopeptidase N; HPRT: hypoxanthine–guanine phosphoribosyltransferase; I/R: ischaemia-reperfusion. Original western blots are presented in Appendix A.

**Figure 6 pharmaceutics-16-00542-f006:**
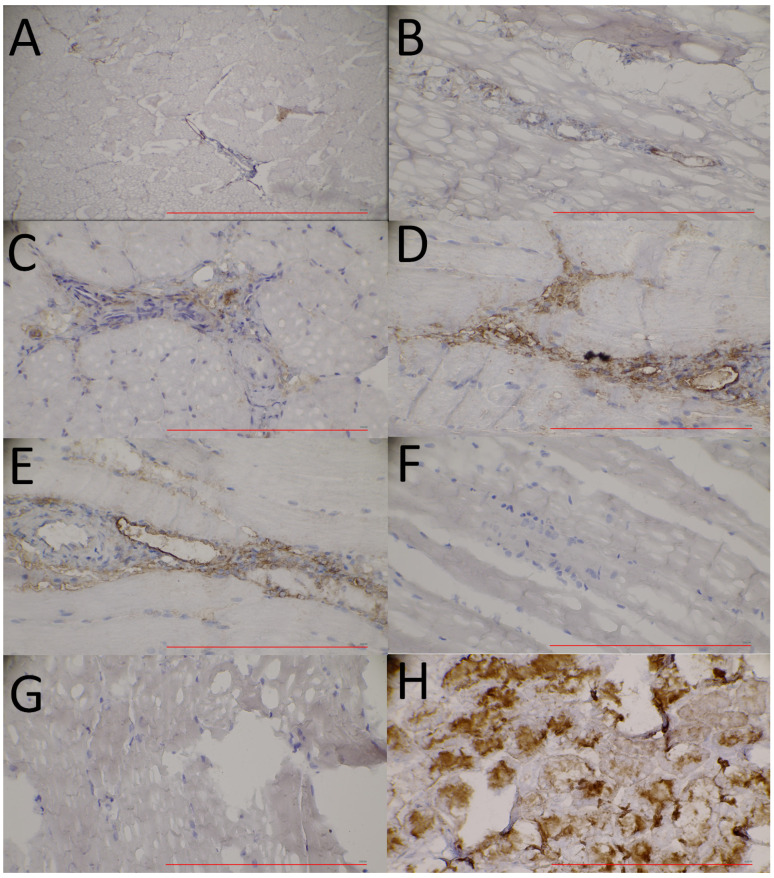
Immunohistochemical analysis of APN/CD13 expression in the ischaemic hindlimb muscle tissue of the ischaemic (I/R) animal group 1 (**A**), 3 (**B**), 5 (**C**), 7 (**D**), and 10 (**E**) days after tourniquet-induced ischaemia. Normally perfused hindlimb muscle tissue samples (non-I/R) of the ischaemic (**F**) and control (**G**) groups were used as negative controls. Rat kidney tissue (**H**) was used as a positive control for APN/CD13 expression. Anti-CD13 primary antibody was visualized with 3,3-diaminobenzidine (DAB) (brown staining). Magnification: 40×. Bar: 200 µm. APN/CD13: aminopeptidase N; DAB: 3,3-diaminobenzidine; I/R: ischaemia/reperfusion.

## Data Availability

Data are contained within the article and Appendix A.

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
