# Peer review of "In Vivo Imaging of Acute Hindlimb Ischaemia in Rat Model: A Pre-Clinical PET Study"

_pharmaceutics, 2024, doi:10.3390/pharmaceutics16040542_

Round 1

Reviewer 1 Report

Comments and Suggestions for Authors

The article address the important topic of acute hind limb ischemia relating PET result with molecular markers of angiogenesis. 

Major comments

Some methodological and ethical aspects must be clarify. 

1. During the 120 min and during reperfusion, did the animals recieved any type of analgesic and/or anesthetic treatment?

2. Were and how the samples were taken, did the animals were under anesthesia?

3. Sex of the animals were not indicated.

4. Sample calculation was missing.

5. A quantification of angiogenesis in the samples should be included.

6. All figures should disclose the number of individuals instead of bars that occult the size of the results.

7. The discussion must be improved specially on a potential explanation of the drop seen at day 10.

Author Response

Dear Reviewer 1;

Thank you for your precious review and valuable comments made on our present manuscript. Please find our answers to your questions and comments below. All changes and supplementations made in the main text could be followed with track changes as requested by the editor of MDPI.

Comment 1. During the 120 min and during reperfusion, did the animals recieved any type of analgesic and/or anesthetic treatment?

Response: In order to prevent the suffering of experimental animals, during the 120-minute-long ischemic period rats were under isoflurane-induced anaesthesia (3% isoflurane), moreover, Ibuprofen (Children’s Motrin) was administered - postoperatively as analgesics - in the drinking water (3.75 ml Ibuprofen (100mg/5ml) to 500 ml water = 0.15mg/ml solution.  This solution provides a dose of 15 mg/kg assuming a 300 gram weighted rat drinks 30 ml/day).

Comment 2.  Were and how the samples were taken, did the animals were under anesthesia?

Response: Experimental animals were under isoflurane-induced anaesthesia during the sampling, and the procedure took place under sterile conditions in the operating room of our department.

Comment 3. Sex of the animals were not indicated.

Response: Male rats were used for the experiments. The “2.2. Animal models” part of the “2. Materials and Methods” section contains the sex of the animals.

Comment 4. Sample calculation was missing.

Response:

The number of animals enrolled in the present study was calculated on the basis of previous studies dealing with hindlimb ischaemia models. In order to make our results comparable with those of earlier findings and to generate clinically translatable conclusions, we intended to apply similar number of animals to former experiments. In addition, to determine the appropriate number of animals we used the so-called G*Power software.

Comment 5. A quantification of angiogenesis in the samples should be included.

Response: The only quantification method applied in the current study was immunohistochemistry for the assessment of APN/CD13 expression. As it was a proof-of-concept study to evaluate the suitability of an NGR-based tracer in ischaemia-reperfusion detection, we preassumed it was enough to use histological staining for angiogenesis quantification. Nevertheless, valuable methods are available for the in-depth in vivo evaluation of angiogenesis, such as the measurement of blood hemoglobin level, or counting the number of newly formed blood vessels using a stereomicroscope. Based on literature data, in hindlimb ischaemia neovascularization is characterized by aggravated endothelial cell proliferation and capillary density (Couffinhal et al., 2009). Therefore, the measurement of capillary density could also be a promising to assess angiogenesis. Although, as muscle atrophy induced by hindlimb ischaemia is a frequent phenomenon in the low oxygenated extremities, calculating the number of capillaries per muscle fiber could be more appropriate than the measurement of capillaries per square millimetre (Emanueli et al., 2001). In addition, among the in vitro techniques, endothelial cell proliferation assays like the determination of net cell number with hemocytometer applying a light microscope or an electronic counter, or the assessment of cell-cycle kinetics could be proper methods for the assessment of higher endothelial proliferation related to angiogenesis (Tahergorabi et al., 2012). Besides these, endothelial cell migration and cell differentiation assays could also aid the investigation of angiogenesis related to ischaemia.

Based on this brief insight into angiogenesis quantification and given that a more profound assessment of ischaemia-related angiogenesis would open ways to discover novel therapeutic and diagnostic targets, the lack of the further measurement of angiogenesis constitutes a significant limitation of the present work.  

References:

Couffinhal T, Dufourcq P, Barandon L, Leroux L, Duplaa C. Mouse models to study angiogenesis in the context of cardiovascular diseases. Front Biosci . 2009;14:3310–3325.

Emanueli C, Minasi A, Zacheo A, Chao J, Chao L, Salis MB, et al. Local delivery of human tissue kallikrein gene accelerates spontaneous angiogenesis in mouse model of hindlimb ischemia. Circulation. 2001;103:125–132.

Tahergorabi Z, Khazaei M. A review on angiogenesis and its assays. Iran J Basic Med Sci. 2012 Nov;15(6):1110-26. PMID: 23653839; PMCID: PMC3646220.

Comment 6. All figures should disclose the number of individuals instead of bars that occult the size of the results.

Response: In the case of all figures (where bars represented) number of individuals were defined in the figure description.

Comment 7. The discussion must be improved specially on a potential explanation of the drop seen at day 10.

Response: The discussion has been supplemented as follows:

“The drop in protein expression observed at day 10, subsequent to the initial increase from day 1 to day 7, may stem from various factors inherent to the dynamic nature of neo-angiogenesis and tissue repair following ischemic injury. One plausible explanation pertains to the resolution of the acute inflammatory response typically associated with tissue injury and ischemia. As the initial insult wanes, there could be a diminution in the expression of pro-inflammatory mediators, leading to reduced recruitment and activation of inflammatory cells, notably macrophages expressing APN/CD13.”

Reference:

Nahrendorf M, Pittet MJ, Swirski FK. Monocytes: protagonists of infarct inflammation and repair after myocardial infarction. Circulation. 2010 Jun 8;121(22):2437-45. doi: 10.1161/CIRCULATIONAHA.109.916346. PMID: 20530020; PMCID: PMC2892474.

We would like to thank you again for your detailed and benevolent review. We hope you will accept our answers and corrections.

Waiting for your positive evaluation,

Yours sincerely,

Gergely Farkasinszky

corresponding author

Reviewer 2 Report

Comments and Suggestions for Authors

The research paper presents a compelling case for the potential of [68Ga]Ga-NOTA-c(NGR) in assessing alterations in APN/CD13 induced by ischemia in a preclinical model of hindlimb ischemia, akin to human PAD. The designed experiments showcased in the paper elucidate this potential with clarity and precision. Additionally, the inclusion of 2-[18F]FDG PET scans to evaluate metabolic changes associated with ischemia further enriches the comprehensive nature of the study. The introduction sets a solid foundation by clearly delineating the research objectives and context. The flow of the manuscript was good, and conclusions were made with enough supportive results. In my opinion, the manuscript can be accepted for publication in Pharmaceutics.

Author Response

Dear Reviewer 2;

Thank you for your precious review and valuable opinion made on our present manuscript.

Yours sincerely,

Gergely Farkasinszky

corresponding author

Reviewer 3 Report

Comments and Suggestions for Authors

The review is quite relevant and appropriate. It reports interesting findings on the feasibility of 68Ga-NOTA in assessing ischemia induced in preclinical hindlimb rat model. 18FDG was also investigated to assess ischemia-metabolic related changes in the same model. The paper is nicely written, presents high quality figures, nice layout and easy flow. However, the viewer has encountered some areas that can be improved before accepting this work for publication. Enclosed below are some general and specific comments for the authors to consider.

General comments

1. How did you distinguish between the signal attributed to 68Ga and 18FDG?  

2. Acknowledging the magnitude of higher FDG uptake comparing to 68Ga, how can this be managed, corrected, compensated during overall assessment?

3. Although limitation associated with each study was mentioned, it might be easier for readers to see all study encountered limitation in one paragraph before the conclusion.

Specific comments:

1.Line 62: spell out the acronym NGR

2. Line 86: Full citation of Mikecz et al is missing

3. Line 115: for consistency keep hind limb one word throughout the paper

4. Line:119-120: more clarification is need on how the control group was determined. Tourniquet was used for all 15 rats or only 10? If only 10, what is the meaning of randomly categorized?

5. Line 140: a typo in density unit!

6. Line 153: spell out the acronym SDS

7. Fig 2 panel D: the error bars represent the deviation across the 10 rats? If yes, why such discrepancies? What are the 4 bins per day?

8. Fig 3 panel D: less discrepancy is seen between the rats in the different day, why?

9. Fig 4: it seems FDG shows no sine of ischemia in the control group! Interestingly, the spatial distribution of both tracers with the VOI is different. Could you elaborate!

10. Line 80: a typo in ischaemia.

Author Response

Dear Reviewer 3,

Thank you for your precious review and valuable comments made on our present manuscript. Please find our answers to your comments below. All changes and supplementations made in the main text could be followed with track changes as requested by the editor of MDPI.

General comments

  1. How did you distinguish between the signal attributed to 68Ga and 18FDG?  

Response: PET examinations were performed separately for both radiotracers ([68Ga]Ga-NOTA-c(NGR) and 2-[18F]FDG). In all cases, we started the PET examinations with the shorter half-life radiotracer and waited for the necessary wash out time, which was at least 6 hours due to the half-life of 68Ga (approx. 68 minutes).

  1. Acknowledging the magnitude of higher FDG uptake comparing to 68Ga, how can this be managed, corrected, compensated during overall assessment?

Response: During the study, we did not compare the results of the different radiotracers ([68Ga]Ga-NOTA-c(NGR) vs 2-[18F]FDG), but rather the values measured on different time points (day 1, 3,5,7,10) with the same radiotracer during the longitudinal study. The results of the examinations with the two radiotracers were described in a separate figure and paragraph.

  1. Although limitation associated with each study was mentioned, it might be easier for readers to see all study encountered limitation in one paragraph before the conclusion.

Response: Limitation associated with the present study was elaborated in a separet paragraph before the conclusion as follows:

“Particular attention must be paid to the study limitations arising from the utilization of a small number of animals in the research. In our study, the use of a limited number of animals, albeit common in preclinical research, may result in restrictions on the generalizability of our findings to human populations. Additionally, the translation of these findings to human subjects is required to be carefully considered due to inherent inter-species differences. Furthermore, a notable limitation was presented by the absence of comprehensive immunological panels, as valuable insights into the underlying mechanisms driving observed outcomes could be offered by such assays. Moreover, valuable methods are available for the in-depth in vivo evaluation of angiogenesis, which could be explored in future studies. Priority should be given in future investigations to the inclusion of a broader array of immunological assays to further elucidate the complex interactions between the experimental interventions and the immune system. Despite these limitations, valuable preliminary data was contributed to the field by our study, underscoring the importance of subsequent research efforts aimed at addressing these constraints and expanding upon our findings. “

Specific comments:

  1. Line 62: spell out the acronym NGR

Response: According to the reviewer suggestion the sentence was modified as follows:

Based on phage display technology results, peptides with asparagine-glycine-arginine (NGR) motif specifically bind to APN/CD13 [17].”

  1. Line 86: Full citation of Mikecz et al is missing

Response: We corrected the citation at the „References” section of the manuscript as follows:

Mikecz, P.; Tóth, G.; Horváth, G.; Lehel, S.; Kovács, Z.; Pribóczki, E.; Boros, I.; Miklovicz, T.; Márián, T. Synthesis of radiopharmaceuticals for PET investigations. Orv Hetil 2002, 143, 1240-1242. Hungarian. PMID: 12077905.

  1. Line 115: for consistency keep hind limb one word throughout the paper

Response: We checked and corrected it throughout the text.

  1. Line:119-120: more clarification is need on how the control group was determined. Tourniquet was used for all 15 rats or only 10? If only 10, what is the meaning of randomly categorized?

Response: In case of the sham-operated control group the same procedures were applied like in the ischemic group (such as anesthesia, surgical preparation, monitoring during surgery) but the clamping of the base of the left hindlimb using a tourniquet was not executed therefore ischemia did not occur.

Simple randomisation method was used, when all animals are simultaneously randomised to the treatment groups without considering any other variable.

  1. Line 140: a typo in density unit!

Response: Thank you for your comment, now it’s corrected as follows:

“VOI: SUV=[VOI activity (Bq/mL)]/[injected activity (Bq)/animal weight (g)], assuming a density of 1 g/mL.”

  1. Line 153: spell out the acronym SDS

Response: The sentence was supplemented as follows:

Corresponding to the molecular weight of the target protein, the lysates were separated on 10% Sodium Dodecyl Sulfate (SDS) polyacrylamide gel by electrophoresis (SDS-PAGE) and electrotransferred onto a  polyvinylidene fluoride (PVDF) membrane (Millipore, Burlington Massachusetts, USA).”

  1. Fig 2 panel D: the error bars represent the deviation across the 10 rats? If yes, why such discrepancies? What are the 4 bins per day?

Response: Error bars represent the deviation across different groups (4 bins per day, number of animals=15) such as normally perfused hindlimb of the control group (non-I/R, left), normally perfused hindlimb of the control group (non-I/R, right), ischaemic (I/R) hindlimb of the ischaemic group (left), and normally perfused hindlimb of the ischaemic group (non-I/R, right). In the case of all figures (where bars represented) number of individuals were defined in the figure description.

  1. Fig 3 panel D: less discrepancy is seen between the rats in the different day, why?

Response: There is less discrepancy seen between the rats in the different days in case of SUVmax assessment (panel D) because the mechanism of action of the tracer (2-[18F]FDG). Interestingly if one compares the radiotracer’s SUVmean and SUVmax values and longitudinal trends there are different characteristics could be found. Possible explanation could be the difference resulting from their mechanism of action (2-[18F]FDG: glucose metabolism vs. [68Ga]Ga-NOTA-c(NGR): APN/CD13 receptor expression).

PET metrics include qualitative assessment; regional semiquantitative indices, such as the mean standardized uptake value (SUVmean)(panel C), which is the average of 2-[18F]FDG uptake activity in an area, and the maximum standardized uptake value (SUVmax)(panel D), which represents the pixel with the highest 2-[18F]FDG uptake activity of the volume of interest. In case of 2-[18F]FDG PET there are pixels with high 2-[18F]FDG uptake activity in all cases but SUVmean values (panel C) are more variable and more correlative to the visual assessment shown on the figures (3 panel A and B).

  1. Fig 4: it seems FDG shows no sine of ischemia in the control group! Interestingly, the spatial distribution of both tracers with the VOI is different. Could you elaborate!

Response: As for the control cohort (control group), both hindlimbs (left and right) belonged to the non-I/R subgroup. In case of Fig 4, I confirm that no sign of ischemia since ischemia was not even induced there. The comparison of the spatial distribution of both tracers can lead to the conclusion that there are reasons for the differences between VOI’s. That could be traced back to the following reasons:

-     Different maximum positron energies of the radionuclides used in the labelling
(68Ga: 1.899 MeV, 18F: 0.635 MeV) 

      Differences between the mechanisms of action of the radiotracers (see comment 8)

-     Differences between the uptake values within the affected volume.

  1. Line 80: a typo in ischaemia.

Response: This typo is corrected in the text.

Thank you again for your propitious review. We hope you will accept our answers and corrections.

Waiting for your positive evaluation,

Yours sincerely,

Gergely Farkasinszky

corresponding author

Round 2

Reviewer 1 Report

Comments and Suggestions for Authors

Most of my comments were answered acceptably. 
The manuscript improved from the previous version, but capillary density measurements to assess angiogenesis and the number of capillaries per muscle fiber from the collected images must be included.  This may involve taking more pictures from the tissues and expanding the analysis, but it will better support the interesting results obtained with the in vivo imaging.